# Tuning evolvability via plasmid copy number and regulatory architecture

**Ximing Li** ⓘ & **Andras Gyorgy** ⓘ ✉

Genetic modules are often designed and implemented with inspiration from engineering disciplines. Although this approach can be successful because of the similarities underpinning physical and biochemical systems, it neglects a key factor that affects the performance of living organisms: evolution. Thus, it is crucial to incorporate the impact of inevitable mutations into the design and analysis of genetic modules. Combining computational modeling and in vivo mutagenesis experiments in *Escherichia coli*, we characterize how the interplay of gene dosage via plasmid copy number (PCN) and regulatory architecture affect the phenotypic mutation rate. For example, while greater PCN facilitates the emergence of gain-of-function mutations, it instead curbs the spread of loss-of-function mutations. We further reveal that mutations in the coding region are often masked at the phenotypic level, unlike those occurring in the regulatory region which become more prominent as PCN increases, both when the regulator is expressed constitutively and when it is self-repressed. Together, our results shed light on evolutionary organizing principles and aid the rational design of both evolutionarily stable and highly evolvable biocircuits.

The design of synthetic gene circuits is often inspired by engineering principles[1,2]. For instance, modules that realize sophisticated analog[3] and digital functions[4] can be implemented by borrowing concepts from electrical engineering[5–10], and complex systems can be realized leveraging computer-aided automated design tools[11,12] in a wide array of contexts ranging from biomanufacturing to living therapeutics[13–18]. Unfortunately, the analogy is imperfect, as modules display context-dependent behavior due to a variety of factors, such as competition for shared cellular resources[19,20] and the physical layout of circuits[21]. Representing an even more fundamental difference, while electrical circuits are static elements, genetic components are instead subject to constant changes due to mutations and evolutionary pressure, eventually leading to design failures[2]. Therefore, it is imperative to integrate evolutionary considerations into the design and analysis of synthetic gene circuits[22–25] to fine-tune the balance between (i) evolvability to optimize or innovate novel traits[26–30], and (ii) evolutionary stability to preserve the phenotype in the presence of genetic mutations[31–34].

Synthetic gene circuits are typically deployed on plasmids, offering a convenient platform to modulate their gene dosage via their copy number. These self-replicating extrachromosomal DNA vectors are present in a wide range of bacteria, and provide an avenue for accelerated evolution[35]. Plasmids are most often shared among bacteria via horizontal gene transfer[36] to aid the spread of beneficial evolutionary traits, such as resistance to antibiotics, degradation of toxic chemicals, the ability to synthesize essential amino acids and to undergo anoxygenic photosynthesis[37–40]. Unlike the chromosome, plasmids are generally present at multiple copies in each host, further accelerating evolution through gene amplification[41]: upon mutation, high copy plasmids are likely to be heterozygous, yielding enhanced genetic diversity and plasticity to overcome evolutionary trade-offs[42].

Numerous in vivo mutation systems have been developed in recent years to improve directed evolution by providing continuous genetic diversification[29,43–46] to closely mimic the dynamics in population genetics, where mutations occur stochastically and across generations. Compared with in vitro mutations, in vivo evolution yields (i) mutations that are directly linked to host fitness; (ii) parallel streams of mutations due to the constantly replicating population; and (iii) potentially wider and longer mutation paths[43]. Thus, instead of utilizing pre-diversified genetic variants, we adopt an in vivo system to generate mutations using EvolvR: an enhanced Cas9 nickase fused with

Division of Engineering, New York University Abu Dhabi, Abu Dhabi, UAE. ✉e-mail: andras.gyorgy@nyu.edu

a modified error-prone DNA polymerase which is directed by a single guide RNA (sgRNA) to generate mutations at a constant rate at user-defined locations[46].

Harnessing EvolvR, we implemented a collection of engineered circuits to study the emergence of plasmid-borne mutations. We show that the interplay between gene dosage and regulatory motifs shapes both genotypic and phenotypic heterogeneity, highlighting the value of integrating synthetic gene circuits and in vivo mutagenesis to study evolutionary dynamics. By developing a computational model to simulate mutation, plasmid replication and partitioning over subsequent generations, we show that the number of cells with at least one mutant plasmid increases with PCN, which we then verify experimentally. Following this, we demonstrate that while for gain-of-function mutations the rate at which phenotypic mutations occur depends on the detection limit, for loss-of-function mutations this rate strictly decreases with PCN. Our work further reveals that mutations in the coding region tend to be masked at the phenotypic level, unlike those in the regulatory region which become more prominent as PCN increases. Our results demonstrate that circuit-level features such as gene dosage and regulatory motifs that shape gene expression impact the emergence and spread of mutants at the phenotypic level despite identical mutation dynamics at the DNA level. Thus, our work provides explicit design guidelines to modulate the dominance of mutants within a population, and to incorporate evolutionary considerations into the design of complex synthetic gene circuits.

## Results

### Phenotypic mutation rate is expected to depend on gene dosage

To investigate how the phenotypic mutation rate is impacted by gene dosage, we adopted a modified version of the standard Wright-Fisher model[47,48] to perform stochastic simulations. This framework is commonly used in population genetics, although typically for haploid and diploid organisms. Thus, we needed to slightly modify it to account for the fact that PCN generally exceeds two, corresponding to a multiploid population in the Wright-Fisher model.

Our computational framework (Fig. 1a) has discrete and non-overlapping generations with a constant population size $N$, and each member is replaced in every generation as follows. Let $x_m$ and $x_{wt} = P - x_m$ denote the number of mutated and wild-type plasmids in a given cell, respectively, where $P$ is the PCN. Similarly, let $N_m$ and $N_{wt} = N - N_m$ denote the number of mutant (with at least one mutated plasmid) and wild-type bacteria (those without any mutated variants), respectively. We generate the number of new mutations $\Delta x$ within each cell as follows: each wild-type plasmid becomes mutated with probability $\mu$, modeled as a Poisson process. Next, the numbers of wild-type and mutated plasmids for each cell are updated (STEP 1 in Fig. 1a) as $x'_{wt} = x_{wt} - \Delta x$ and $x'_m = x_m + \Delta x$, respectively. Subsequently, the number of new plasmids $P'$ after replication is generated according to a second Poisson distribution, with its mean equal to the average PCN (see Methods), and this pool is split according to a binomial distribution with parameters $P'$ and $x'_m/x'_{wt}$, that is, newly replicated plasmids are randomly assigned to be either wild-type or mutated according to the respective proportions within the cell (STEP 2 in Fig. 1a). Following this, both daughter cells receive either $(P + P')/2$ plasmids if $P + P'$ is even, otherwise one of them ends up with one more than the other to ensure that there is no segregational loss, where a hypergeometric distribution[49] determines how mutated and wild-type plasmids are distributed (STEP 3 in Fig. 1a). Finally, half of the population is randomly selected to initialize the next generation (STEP 4 in Fig. 1a), while the rest is discarded to keep the population size constant across generations.

Relying on parameter values typical in *E. coli* (see Methods), data in Fig. 1 reveal the following. First, the size of the mutant subpopulation increases with PCN (Fig. 1b), and while the average number of mutated plasmids also increases with PCN within this subpopulation (Fig. 1c), their fraction instead decreases (Fig. 1d). Greater gene dosage thus results in more bacteria with at least one mutated plasmid, and they will harbor more mutated variants in absolute numbers, but not in relative terms. These trends are insensitive to changes in parameter values (Supplementary Fig. 1), and while mutations that arise in early generations tend to dominate the mutant population for low PCN, increasing gene dosage contributes to greater genotypic diversity with mutations emerging across multiple generations (Supplementary Fig. 2).

In what follows, we consider the population-level effects of plasmid mutations, emphasizing how circuit-level properties (e.g., PCN,

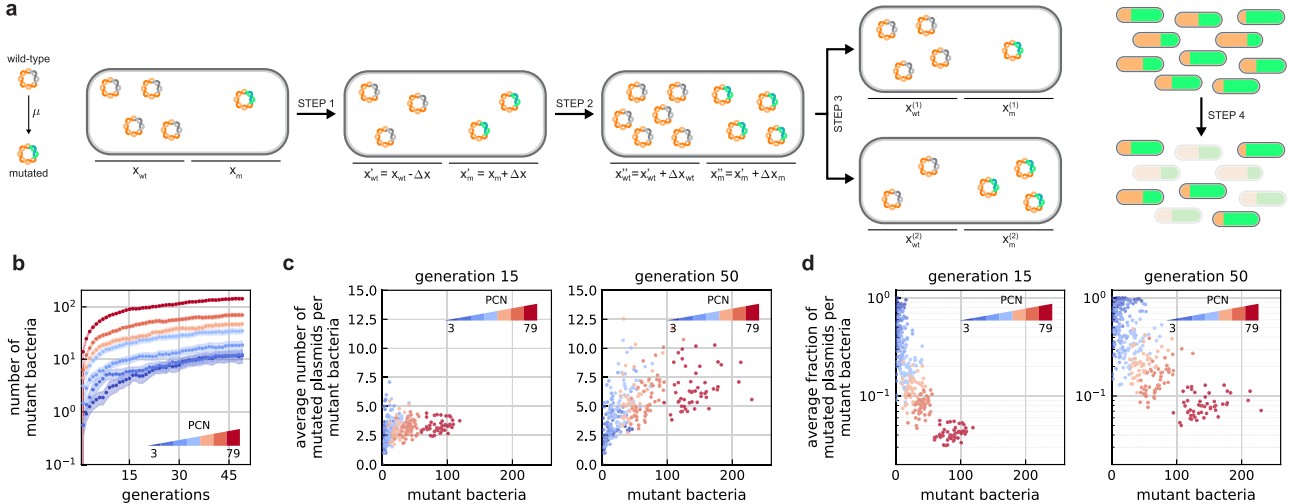

**Fig. 1 | Stochastic simulations reveal that while the number of bacteria harboring mutant plasmids increases with PCN, their prevalence within each cell instead decreases.** Selected values of PCN correspond to the average for the pSC101 variants used in subsequent genetic constructs (i.e., 3, 5, 9, 17, 25, 39, and 79). For each value, the results of 50 independent replicates are presented. **a** Within each generation, first mutations are generated, then plasmids are randomly distributed after their replication. Subsequently, half of the population is selected to initialize the next generation, thus keeping the population size constant across generations. **b** The subpopulation harboring at least one mutated plasmid increases with PCN. For each PCN, the means from 50 independent replicates are presented across generations. Shaded areas represent 95% confidence intervals of the estimated mean values. **c** The number of mutated plasmids per mutant bacteria increases with PCN. **d** The fraction of mutated plasmids per mutant bacteria decreases with PCN.

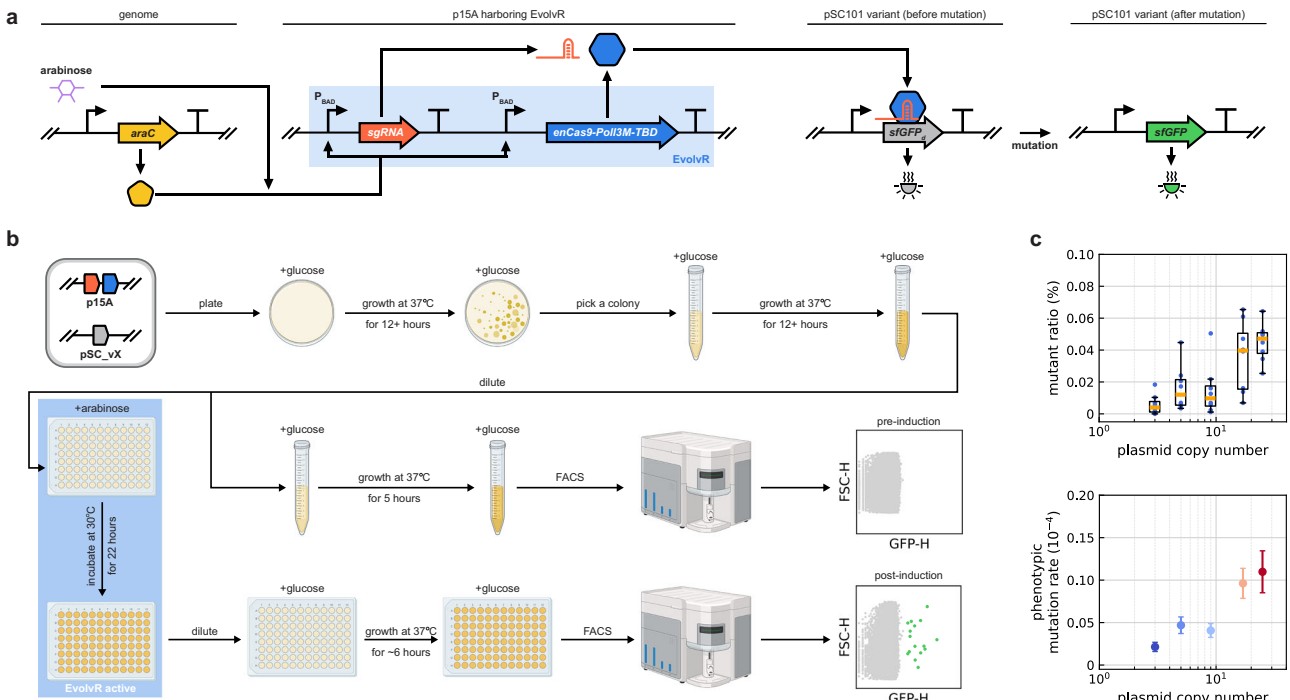

**Fig. 2 | Phenotypic mutation rate depends on gene dosage. a** The inducer arabinose together with AraC expressed from the genome activates EvolvR, which is harbored on a p15A plasmid to generate mutations using an engineered DNA polymerase and Cas9 nickase. The resulting fusion protein (enCas9-PolI3M-TBD) targets a 20 basepair long window at a user-defined location, specified by the sgRNA. A secondary plasmid with a variant of the pSC101 ori contains the non-fluorescent *sfGFP*$_d$ gene. **b** Following co-transformation with both plasmids, cells are induced with arabinose to activate EvolvR. Data in Supplementary Fig. 4 highlight that the presence of arabinose or the synthesis of sfGFP$_d$ do not impact cell growth, unlike the expression of EvolvR. To mitigate this burden, we adopted 30 °C during the induction stage. We allow a flexible regrowth window for post-induction cultures that spans approximately 6 h to ensure that the cultures reach comparable growth states in exponential phase before subsequent FACS analysis for mutant detection. The mutant subpopulation is identified using flow cytometer data (Supplementary Fig. 8), comparing the pre-induced and post-induced samples (Supplementary Fig. 19). Created in BioRender. Gyorgy, A. (2025) https://BioRender.com/3mvgrft. **c** The mutant ratio is obtained in flow cytometer experiments (Supplementary Fig. 8) using eight independent replicates, which is then leveraged using fluctuation analysis[98,99] to derive the phenotypic mutation rate (error bars represent 95% confidence intervals of the estimated mutation probability). See Supplementary Fig. 11 for data on mutant ratios and pairwise statistical significance tests across PCN values. In the boxplot figure, centers indicate the median. The lower and upper bounds of the box represent the first quartile (Q1) and third quartile (Q3), respectively. With IQR = Q3−Q1 denoting the interquartile range, the whiskers extend from the box to the minimum and maximum non-outlier values such that the lower and upper thresholds are defined as Q1−1.5IQR and Q3+1.5IQR, respectively.

type and location of the mutation, regulatory architecture) modulate the phenotypic mutation rate within a population. While not the focus of our study, it is important to note that the co-occurrence of multiple plasmid variants within a single cell can result in sophisticated dynamics due to the interplay of evolutionary forces acting at two scales: in addition to competition between cells, host fitness is also impacted by replication interference between plasmids harbored within the same host. Studying this latter, intracellular competition, is possible via novel experimental tools such as engineered dimer plasmids[50] or via Cas9-based lineage recording[51] to quantify the fundamental trade-off between the two levels of selection, but falls outside the scope of our work.

## Phenotypic mutation rate depends on gene dosage

To experimentally characterize whether and how gene dosage affects the phenotypic mutation rate, we first considered a simple genetic circuit comprising two plasmids (Fig. 2a). The first one with a p15A origin of replication (ori) contains the CRISPR-guided error-prone DNA polymerase EvolvR[46] and sgRNA, both expressed from the arabinose-inducible $P_{BAD}$ promoter. EvolvR is designed to edit within a window extending to around 60 nucleotides in the 3′ direction from the Cas9 nick site[46]. The second plasmid is equipped with a pSC101 variant ori, selected from a set with PCN ranging from 3 to 25 per chromosome[52], and it harbors constitutively expressed sfGFP with a defect, rendering

it non-fluorescent (*sfGFP*$_d$ gene) by changing residue 88 from methionine to a stop codon. Relying on the expression of araC in the genome (Supplementary Fig. 3), synthesis of EvolvR and the sgRNA is induced via the addition of arabinose to target the stop codon region, thus correcting the defect in the *sfGFP*$_d$ gene to regain fluorescence.

After co-transformation with both plasmids, cells were induced by arabinose and sfGFP expression was monitored using flow cytometry (Fig. 2b). This revealed that a small fraction of events (less than 1%) showed varying levels of high sfGFP expression, corresponding to bacteria containing mutated plasmids (with *sfGFP* instead of *sfGFP*$_d$). As expected, EvolvR activity is largely confined to the target editing window, correcting the premature stop codon and restoring function resulting in the dominance of a small subset of genotypes (Supplementary Fig. 5b). In general, the phenotypic mutation rate increases with PCN (Fig. 2c) as the potential targets increase, positively contributing to the mutational supply, eventually leading to a greater mutant subpopulation. Unfortunately, PCN also increases the total concentration of the target protein (sfGFP$_d$ and sfGFP combined) for the circuit featured in Fig. 2a. Since we are interested in how the impact of PCN on the mutational supply alone affects the size of the mutant subpopulation, we next integrated a control module to ensure that the expression of the target protein remains unchanged as we explored how the phenotypic mutation rate depends on the gene dosage both in case of gain-of-function and loss-of-function mutations.

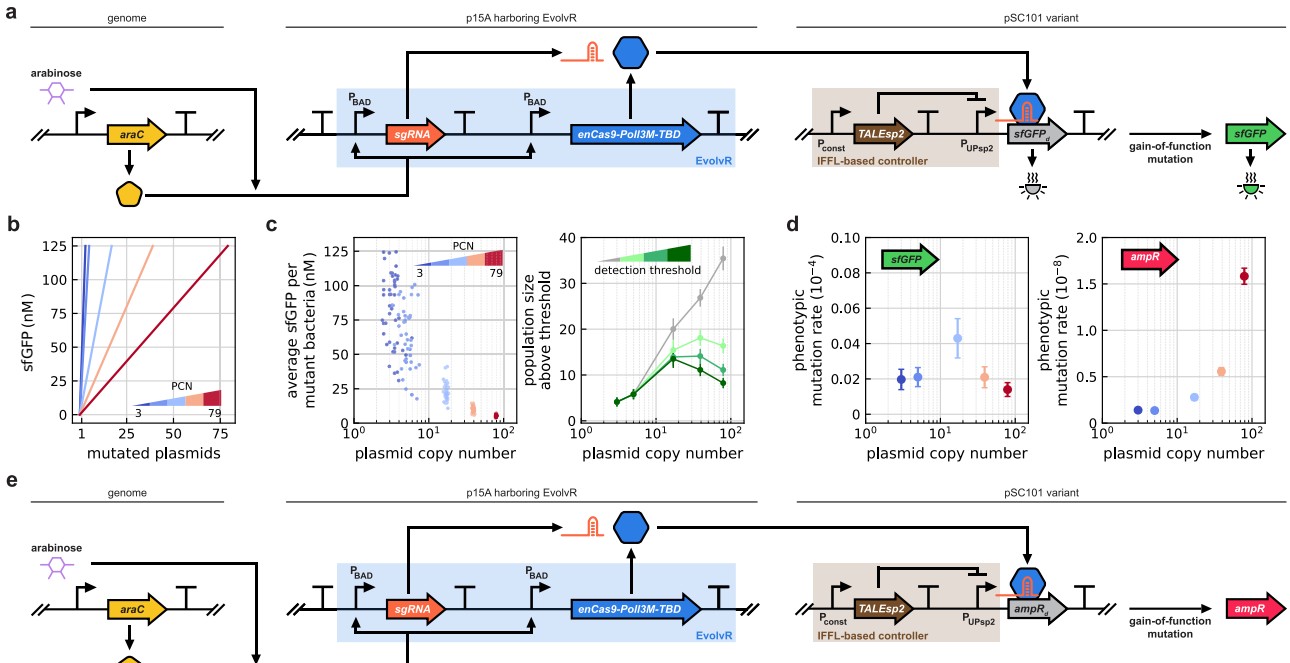

**Fig. 3 | Regulating the synthesis of the target protein impacts the phenotypic mutation rate.** The IFFL-based control module regulates the expression of the target protein to ensure that its synthesis rate is independent of PCN. **a** The target gene is *sfGFP*$_d$ from Fig. 2a. **b** The production rate of fluorescent sfGFP per mutated plasmid decreases with PCN due to the control module, ensuring that the total production of sfGFP (including both sfGFP and sfGFP$_d$) remains constant across pSC101 variants. Data obtained in numerical simulations. **c** The average sfGFP level in mutant bacteria remains a decreasing function of PCN even after the inclusion of the control module. Data obtained in stochastic simulations, considering 50 independent replicates for each PCN. While the size of the detected mutant population increases monotonically with PCN if the detection limit is low (gray), the

relationship can instead become non-monotonic as the threshold increases (green shades). Points represent mean population size and error bars indicate 95% confidence intervals. **d** The phenotypic mutation rate is estimated using fluctuation analysis based on the number of events above the detection limit in flow cytometer experiments when the target is *sfGFP*$_d$ (Supplementary Fig. 20), and based on selective agar plating experiments when the target is *ampR*$_d$[98]. Mean phenotypic mutation rate is obtained from 8 independent replicates and 3 independent trials for the targets *sfGFP*$_d$ and *ampR*$_d$, respectively. Error bars represent 95% confidence intervals. See Supplementary Figs. 12, 13 for data on mutant ratios and pairwise statistical significance tests across PCN values. **e** The target gene is *ampR*$_d$.

## Gene regulation impacts the phenotypic mutation rate

To ensure that the expression level of the reporter protein remains unaffected by variations in PCN, we integrated an incoherent feed-forward loop (IFFL) control circuit[52] to regulate the synthesis of the target protein (Fig. 3a). Instead of constitutively expressing sfGFP (Fig. 2a), its synthesis is driven by the promoter P$_{UPsp2}$, repressed by a transcription activator-like effector (TALEsp2). Changes in PCN affect both TALEsp2 and sfGFP expression, but the impact on the latter can be canceled by fine-tuning the repression of P$_{UPsp2}$ via adjusting the TALEsp2 synthesis, yielding constant sfGFP concentration irrespective of the PCN (Supplementary Fig. 9). Integrating the IFFL-based control module therefore decouples the expression of the target gene from PCN variations, allowing the effects of PCN on mutation to be studied independently.

For a given PCN, the concentration of sfGFP within a cell is expected to increase linearly with the number of mutated plasmids (Fig. 3b). Additionally, the strength of each individual P$_{UPsp2}$ promoter (i.e., the amount of sfGFP expressed, either defective or functional) is now inversely proportional to the PCN due to repression by TALEsp2, as the total amount is independent of the PCN because of the control circuit. Thus, for a given number of mutated plasmids, the amount of functional sfGFP decreases with PCN (Fig. 3b). As the integration of the control module does not impact the mutation events and the simulation results presented in Fig. 1, we expect that increasing gene dosage results in (i) greater mutant subpopulation (Fig. 1b), but (ii) smaller fraction of mutated plasmids within each mutated cell (Fig. 1d). Therefore, two competing factors emerge that shape the population-level behavior: more mutant bacteria are present as PCN increases, but

at the same time, P$_{UPsp2}$ repression also becomes more prominent, decreasing sfGFP production in mutated cells (Fig. 3b). Assuming that the detection limit is low and only non-defective sfGFP produces fluorescence, the latter factor is irrelevant, and we would expect the mutant ratio to increase with PCN (gray in Fig. 3c). However, for any non-zero threshold, the initial increase in the mutant ratio may be followed by a subsequent decrease, as typically there are only a handful of mutated plasmids per mutated cell and the amount of sfGFP they each produce decreases with PCN due to the control module. This is precisely what emerges both in stochastic simulations (Fig. 3c) and in experimental data (Fig. 3d).

To further test the model prediction when the detection limit is low, we next replaced the *sfGFP*$_d$ gene in the circuit outlined in Fig. 3a with a defective ampicillin resistance marker (*ampR*$_d$), as even low expression of an antibiotic resistance gene is expected to ensure survival in selective media. The mutated marker was obtained by changing two consecutive residues (104 methionine and 105 serine) with stop codons to minimize possible ribosomal read-through, which could lead to unintended survival in selective media. Expression of this gene remains regulated by the IFFL-based control module, and the sgRNA was redesigned to target the defect-causing stop codons (Fig. 3e). After induction with arabinose to activate EvolvR targeting these stop codons, cultures were plated on carbenicillin selective agar, then survival colonies were counted to estimate the rate of gain-of-function mutations. Fluctuation analysis reveals that the phenotypic mutation rate sharply increases with PCN (Fig. 3d), confirming the model prediction for the low detection threshold case (gray in Fig. 3c).

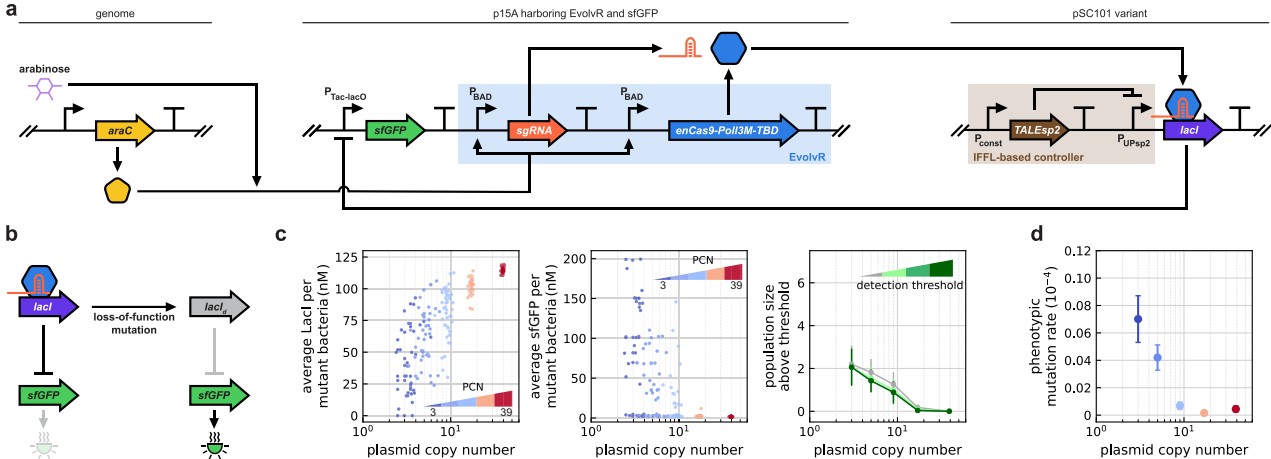

**Fig. 4 | Phenotypic loss-of-function mutation rate decreases with gene dosage.**
The IFFL-based controller regulates the expression of the target protein to ensure that its synthesis rate is independent of PCN. **a** The fluorescent reporter gene *sfGFP* is repressed by LacI, which is both regulated by the controller and targeted by EvolvR. **b** The mutated version of LacI fails to bind to $P_{\text{Tac-lacO}}$, thus relieving the repression. While off-target mutations become more prevalent when targeting *lacI* instead of *sfGFP*$_d$, the majority of these unintended mutations are still restricted to the *lacI* gene, giving rise to the same mutation phenotype (Supplementary Fig. 5). **c** Since the average LacI concentration per mutant bacteria increases with PCN, the average sfGFP level decreases. The size of the detected mutant subpopulation

decreases with PCN independent of the detection limit. Data obtained in stochastic simulations. Points represent mean values from 50 independent replicates and error bars indicate 95% confidence intervals. **d** The phenotypic mutation rate is estimated using fluctuation analysis based on the number of events above the detection limit in flow cytometer experiments (Supplementary Fig. 21) using DH5α∆lacI::ampR cells. Points represent mean values from eight independent replicates and error bars indicate 95% confidence intervals. See Supplementary Fig. 14 for data on mutant ratios and pairwise statistical significance tests across PCN values.

## Phenotypic loss-of-function mutation rate decreases with gene dosage

To reveal how the phenotypic mutation rate changes with gene dosage for loss-of-function mutations, we next included a LacI-repressible *sfGFP* cassette using the $P_{\text{Tac-lacO}}$ promoter on the plasmid with a p15A ori carrying EvolvR (Fig. 4a). Together with the control module regulating its expression, the *lacI* gene is harbored on the other plasmid with a pSC101 variant ori. Arabinose induction activates EvolvR, and by targeting the DNA-binding domain of *lacI* with the sgRNA yields its mutated variant LacI$_d$. This protein either misfolds or loses its binding affinity to $P_{\text{Tac-lacO}}$, thus interfering with its inhibitory function on sfGFP expression (Fig. 4b). The total concentration of the regulator (LacI and LacI$_d$ together) modulating the expression of the fluorescent reporter is independent of PCN due to the control module.

For a given PCN, LacI$_d$ is expected to increase linearly with the number of mutated plasmids (as sfGFP did in Fig. 3b), with decreasing sensitivity as the gene dosage increases due to the control module. Since the fraction of mutated plasmids per mutated cell decreases with PCN (Fig. 1d), more LacI will remain functional (Fig. 4c), mitigating the mutation impact and yielding a downward pressure on sfGFP expression (Fig. 4c). Thus, while the number of mutated bacteria increases with PCN, they each express negligible amounts of sfGFP as the non-mutated *lacI* genes can compensate, hence the population size above the detection threshold is expected to decrease with PCN (Fig. 4c), confirmed experimentally in Fig. 4d.

## Mutations in the coding region are masked at the phenotypic level, whereas those in the regulatory region become more prevalent as PCN increases

Autoregulated motifs play a fundamental role in modulating gene expression and shaping evolutionary adaptation[10,53–58]. Given their wide-spread nature in both natural systems and synthetic biology applications, we next focused on the phenotypic mutation rate in the presence of this central motif. Additionally, since the behavior of a gene circuit may be compromised due to mutations that occur in either the coding sequence of a regulator or its cognate binding site[59,60], we designed sgRNAs that target either of these sites.

In particular, we first implemented sfGFP expression from a promoter that is repressed by LacI, which is synthesized constitutively (Fig. 5a). Both of these genes are harbored on the same plasmid equipped with an ori selected from the set of pSC101 variants to explore the impact of changes in PCN. Next, we placed *sfGFP* and *lacI* in a polycistronic configuration within the same operon (Fig. 5b) to implement negative feedback. In both of these cases, the EvolvR-associated sgRNA targets either the DNA binding domain of the *lacI* gene or its cognate promoter.

When LacI is expressed constitutively from the pSC101 variants (Fig. 5a), its concentration increases with PCN, whereas sfGFP synthesis is governed by two opposing forces. While greater PCN results in an upward pressure due to the increased gene dosage of the *sfGFP* cassette, its activity is also downregulated due to the elevated prevalence of the repressor LacI. While mutating either the *lacI* gene or its cognate promoter results in elevated sfGFP expression and thus in the emergence of a mutant subpopulation, the way this is achieved is fundamentally different, yielding considerable discrepancies in the phenotypic mutation rate and how it depends on PCN. In particular, mutating the *lacI* gene results in protein misfolding or impaired binding. However, other plasmids with intact *lacI* can easily compensate, hence mitigating or altogether eliminating the overall impact of the mutation. Furthermore, this compensation effect is expected to increase with PCN as LacI becomes overabundant, thus saturating the available binding sites and sharply decreasing the average sfGFP per mutant bacteria. This in turn makes it difficult for mutations to manifest at the phenotypic level when the coding region is targeted (Fig. 5a), as the number of wild-type variants that are ready to compensate increases with PCN (Fig. 1). Conversely, mutations in the binding site that prevent its repression via LacI will not be masked by intact copies of the promoter: even a single copy with compromised binding affinity to LacI will express sfGFP independent of the other copies, thus manifesting the mutation at the phenotypic level. Greater PCN increases the mutational supply, which in turn yields increased phenotypic mutation rate when the promoter is targeted (Fig. 5a).

Note that the distribution of mutated cells and mutated plasmids within each cell are independent of the genetic layout, i.e.,

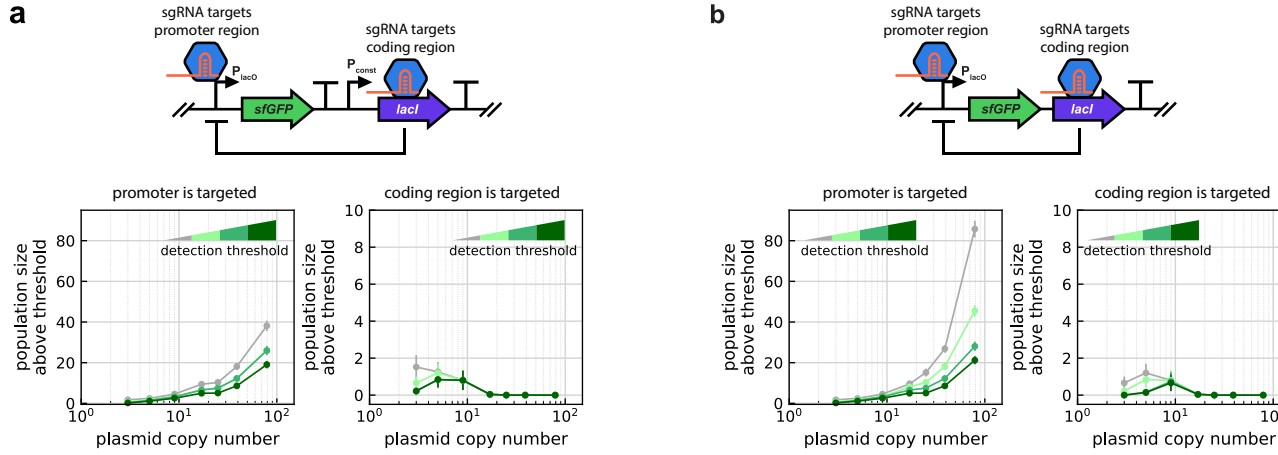

**Fig. 5 | Numerical simulations predict that mutations in the coding region are masked at the phenotypic level, whereas those in the regulatory region become more prevalent as PCN increases.** EvolvR can introduce mutations in either the coding region of *lacI* or its cognate promoter, thus relieving the repression of sfGFP expression. See Supplementary Section 5 for the deterministic models that underpin the simulation data. Points represent mean values from 50 independent replicates and error bars indicate 95% confidence intervals. **a** Expression of LacI is constitutive. **b** Expression of LacI is under negative autoregulation.

whether LacI is expressed constitutively (Fig. 5a) or repressing its own synthesis (Fig. 5b). As a result, the observations uncovered in Fig. 1 hold for both circuits in Fig. 5: the number of mutated hosts increases with PCN, but the fraction of mutated plasmids within each cell decreases. Therefore, the phenotypic mutation rate is expected to follow the same trend when *lacI* is self-repressed (Fig. 5b) as in the constitutive case (Fig. 5a).

To confirm these predictions experimentally, we implemented both circuits outlined in Fig. 5 on various pSC101 variant plasmids (each with a different PCN), and co-transformed them with a p15A plasmid harboring EvolvR alongside with the sgRNA targeting either the *lacI* coding region or its cognate promoter (Fig. 6a). Additionally, we integrated a TetR-based repression module into the two-plasmid system to repress both LacI and sfGFP expression during EvolvR activation. We included this additional layer of control as Cas9-based systems are known to suffer from off-target effects, especially when binding sites become saturated due to the high relative prevalence of Cas9 with respect to its target[61–63]. This likely occurs when the sgRNA targets the promoter, as LacI binding to *lacO* blocks EvolvR from accessing the promoter[64], diverting EvolvR to unintended sites and resulting in off-target mutations (Supplementary Fig. 6). By modifying the promoters of the circuits in Fig. 5 to include additional repression via TetR, the transcription of *sfGFP* and *lacI* are repressed upon arabinose induction both when *lacI* is expressed constitutively (Fig. 6b) and when it is self-repressed (Fig. 6c). Data in Fig. 6bc confirm the theoretical predictions underpinned by the simulations in Fig. 5: mutations in the coding region are masked at the phenotypic level, whereas those in the regulatory region become more prevalent as PCN increases.

## Discussion

Mutations that operate at the DNA level inescapably impact performance at the phenotypic level. While seemingly equivalent regulatory motifs can realize identical functionalities, they may have fundamentally different effect on population structure, growth dynamics, and evolutionary outcomes[56], for instance, based on the demand for the target gene product[65–69]. Furthermore, by sculpting the evolutionary landscape, circuit topology can shape the distribution of mutated variants by imposing evolutionary constraints on potential phenotypes[70,71]. Therefore, it is imperative to understand how circuit-level features link genetic mutations to the spread of mutants in the presence of inevitable evolutionary forces to uncover the organizing principles of living cells, and to devise design guidelines that ensure the predictable behavior of engineered biocircuits. This is especially crucial when evolving genetic circuits at an intermediate scale between focusing on optimizing single genetic components and facilitating genome-wide adaptations in serially propagated cell populations[72].

Using both experiments and stochastic simulations, we characterized how mutations exhibit distinct phenotypic effects as a result of the interplay between gene dosage and regulatory motifs that shape gene expression. For instance, without the IFFL-based control module, the phenotypic mutation rate increases with gene dosage due to the greater mutational supply[35,73]. This in turn could facilitate the acquisition of antibiotic resistance[41,74] or the invention of de novo protein functions[75]. Of course, these potential evolutionary benefits come at a price, as carrying high copy plasmids often results in considerable metabolic burden[76,77]. Integrating an additional control layer (like the IFFL-based regulatory module or negative autoregulation) can relax such bioenergetic constraints[56], allowing the circuits to persist on high copy plasmids. This in turn can directly impact their evolvability among other benefits[57,58], likely contributing to the ubiquitous nature of these motifs in transcription networks[78,79].

The effects of PCN on phenotypic mutations align with the concept of genetic dominance. Although the term is usually reserved to describe the relationship between a diploid genotype and the observed phenotype, different levels of dominance can arise as a result of gene regulation and interactions between plasmids when multiple copies are present[60]. Gain-of-function mutations are often considered dominant, whereas loss-of-function mutations are instead regarded as recessive[80]. Furthermore, gene dosage is often positively correlated with dominant mutations, and negatively with recessive ones[35]. This is confirmed by our results: the phenotypic mutation rate generally increases for gain-of-function or dominant mutations with PCN, and conversely, decreases for loss-of-function or recessive mutations. Additionally, inclusion of a control layer can modulate the mapping from genotypic mutations to phenotypic mutation rate depending on the detection threshold, which could be crucial when establishing genetic dominance[81]. That is, by regulating gene expression, we can adjust the visibility and significance of inevitable mutations, effectively altering the dominance of genetic traits[81].

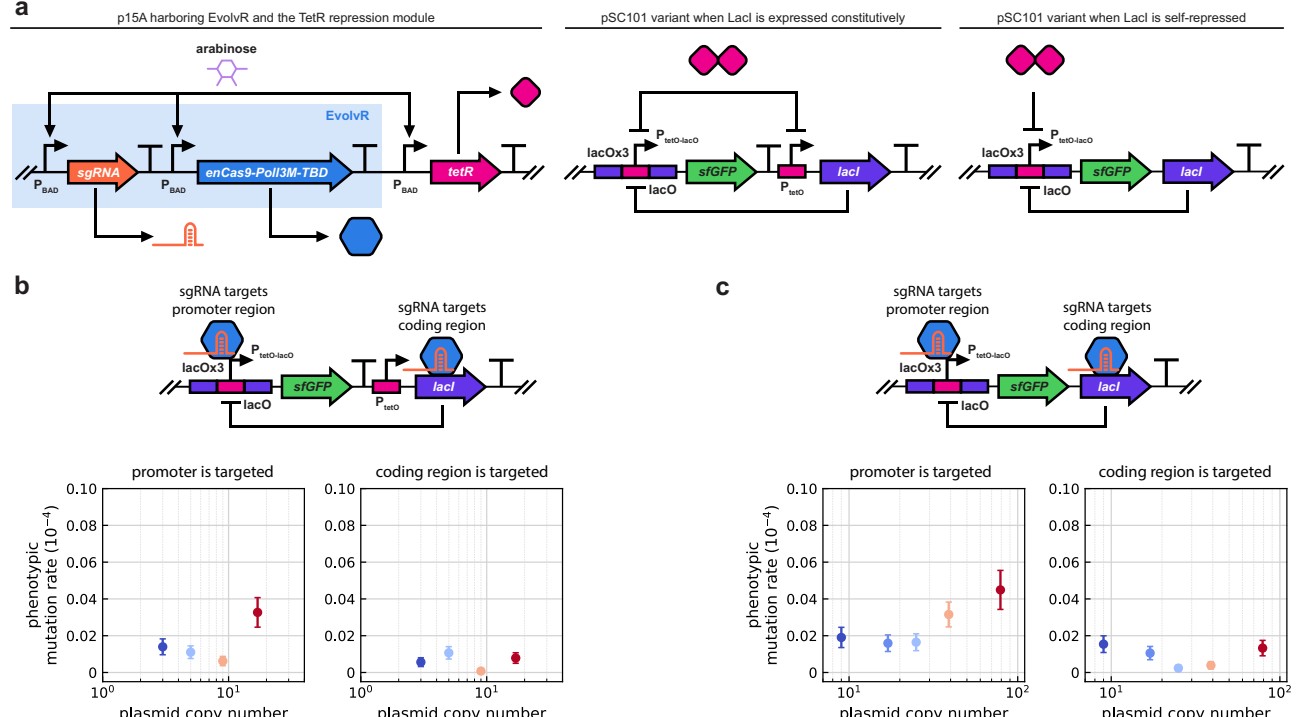

**Fig. 6 | Experimental data confirm that mutations in the coding region are masked at the phenotypic level, whereas those in the regulatory region become more prevalent as PCN increases.** DH5α∆lacI::ampR cells are co-transformed with a p15A plasmid and a pSC101 variant. Arabinose induction is reduced from 2 mM to 200 $\mu$M to decrease the activity of EvolvR and the chance of its binding to off-target sites. Subsequent to arabinose induction, anhydrotetracycline (aTc) at a final concentration of 100 ng/mL is supplemented to relieve the effects of residual TetR. The phenotypic mutation rate is estimated using fluctuation analysis based on the number of events above the detection limit in flow cytometer experiments (Supplementary Figs. 22–25). **a** To prevent leaky expression of sfGFP, a *lacO* array containing three LacI binding sites (*lacO*x3) is placed upstream to the P$_{tetO\text{-}lacO}$ promoter to increase the effective local concentration of LacI[101–103]. **b** Expression of LacI is constitutive. Cells harboring pSC101 variants with

PCN at least 25 become unstable, likely as a result of the metabolic burden due to the synthesis of LacI (Supplementary Fig. 10). Points represent mean values from 8 independent replicates and error bars indicate 95% confidence intervals. See Supplementary Figs. 15, 16 for data on mutant ratios and pairwise statistical significance tests across PCN values. **c** Expression of LacI is under negative autoregulation. On-target mutations are found at a few specific positions that are functionally important, disrupting the *lacO* sites, resulting in the dominance of a small subset of genotypes (Supplementary Fig. 6a). Points represent mean values from 8 independent replicates, and error bars indicate 95% confidence intervals. See Supplementary Figs. 17, 18 for data on mutant ratios and pairwise statistical significance tests across PCN values (even with the inclusion of the TetR repression module and the reduced EvolvR expression, off-target mutations persist for pSC101 variants with PCN below 9, thus the corresponding data are omitted here).

Mutations in the promoter region play a central role in tuning expression levels and rewiring regulatory networks[59,82]. Furthermore, they are generally considered co-dominant[83], whereas those in coding sequences are often regarded as recessive. This is further echoed by our findings: targeting the promoter region with EvolvR yields a greater phenotypic mutation rate than introducing errors in the transcription factor itself, and this differential effect is preserved even in the presence of self-repression. Importantly, EvolvR activity is mostly confined to the gene of interest (though not necessarily to the editing window), causing virtually no genomic mutations, only occasionally spilling over into other elements of the plasmid. Our data also highlight that besides introducing an additional control layer (such as our TetR-based repression module) or reducing EvolvR expression, increasing the abundance of the target site via PCN control can offer an effective alternative strategy to combat this undesirable characteristic of Cas9-based tools.

Synthetic biology research increasingly emphasizes the evolutionary aspect of living organisms as a critical component to successfully implement complex biosystems[22,24]. Bacterial plasmids represent an inherently evolvable platform that can be harnessed in a variety of settings, for instance, to increase the evolutionary stability of burdensome synthetic circuits or to stabilize and contain the function of engineered populations within their target environments[32–34], as well as to engineer in vivo evolution systems with an enhanced capability of

genetic diversification[84,85]. Beyond these systems, the role and impact of intrinsic plasmid heterogeneity can be further studied using advanced experimental techniques such as induced plasmid monomerization[50] or via Cas9-based lineage recording[51] to genetically encode plasmid replication events. While directed evolution is widely exploited to optimize proteins and genetic circuits[30], the reciprocal effects of genetic circuits on evolution itself are considerably less explored. Our work provides explicit design guidelines by revealing how regulatory architecture and gene dosage via PCN together shape the emergence of mutants within a population, despite the simplicity of our computational model (e.g., mutations can be neutral or even increase protein stability and enhance binding[86,87]) and the imperfect nature of the mutation platform (e.g., off-target effects of EvolvR[46]). Therefore, the results presented here shed light on evolutionary driving forces, and can be leveraged to guide the design of evolutionarily stable synthetic gene circuits or to dynamically program genetic diversification by leveraging increased evolvability[84,85], for instance, via flexible PCN control[77,88].

## Methods

### Parameter values of the modified Wright-Fisher model

The mutation rate $\mu$ depends primarily on the concentration of the EvolvR fusion protein and the sgRNA, as well as on the kinetics of DNA binding. Considering prior estimates[46], we chose $\mu = 2.5 \cdot 10^{-6}$

mutations per nucleotide per generation. The constant population size was selected to be $N = 10^5$ in all simulations. In our experimental setup, cells are grown at 30 °C for 22 h with arabinose induction, which corresponds to approximately 15 generations given that *E. coli* doubling time is about 90 min at this temperature[89]. Parameter values of the ordinary differential equations governing the deterministic dynamics of each circuit are provided in Supplementary Sections 4, 5.

## Stochastic plasmid replication

At each generation, new plasmids are produced in each cell following a Poisson distribution with the mean equal to the PCN of the corresponding pSC101 variant, as has been previously developed for R1 plasmids[90] since both systems use RepA-mediated plasmid replication to control PCN. While the specifics of the replication mechanisms differ in R1 and pSC101 plasmids (co-regulation of RepA synthesis via CopA and CopB in the former, and negative autoregulation of RepA expression in the latter), they are both underpinned by the same feature: replication commences when sufficient RepA initiator protein binds to the iterons within the origin of replication, and the process is regulated via the repression of RepA to stabilize PCN.

## Bacterial strains

Clonings were performed using *E. coli* NEB Stable (C3040H). Experiments in Figs. 2, 3 were carried out using DH5α from NEB (C2978H), while those in Figs. 4 and 6 were performed using DH5αΔlacI::ampR. To generate DH5αΔlacI::ampR, we replaced the genomic *lacI* gene in DH5α with an *ampR* cassette using homologous recombination via the λ-Red system[91], following the protocol described in ref. 88. The pRE-DCas9 plasmid carrying the λ-Red was a gift from Tao Chen (Addgene plasmid #71541)[92]. The genome sequence was confirmed via whole genome next generation sequencing (NGS) (Novogene Singapore). Detailed genotypes for the strains used in the study are specified in Supplementary Table 2.

## Media and growth conditions

LB media was prepared using premix powder from Sigma Aldrich (L3522). Antibiotics were sourced from Fisher Scientific: kanamycin sulfate (10031553), chloramphenicol (10324980) and carbenicillin (10396833). Final working concentrations are 50 μg/mL for kanamycin, 25 μg/mL for chloramphenicol, and 100 μg/mL for carbenicillin. L-arabinose was obtained from Sigma Aldrich (A3256), and its final working concentration is 2 mM for experiments in Figs. 2–4 and 200 μM for those in Fig. 6. Anhydrotetracycline (aTc) was obtained from IBA Lifesciences (2-0401-002) with final working concentration of 100 ng/mL.

## Cloning and plasmid assembly strategies

The plasmid harboring the EvolvR gene is constructed based on pEvolvR-enCas9-PolI3M-TBD[46] (Addgene plasmid #113077). The IFFL control circuit (Addgene plasmid #109254) together with the set of pSC101 variants (Addgene plasmids #109240–#109246) are adopted from ref. 52. All plasmids are built using standard molecular cloning procedures[93], including PCR amplification, restriction digestion, gel electrophoresis, ligation, and transformation. Primers and gene blocks were synthesized by Integrated DNA Technologies. All constructed plasmids in the study were confirmed by either Nanopore sequencing (Plasmidsaurus) or NGS (Sangon Biotech). A comprehensive list of all plasmids and DNA parts can be found in Supplementary Tables 3, 4, together with the plasmid maps in Supplementary Fig. 26.

## Transformation procedure

Heatshock transformation was carried out using chemically treated NEB Stable competent cells for cloning, and *E. coli* DH5α or DH5αΔlacI::ampR for experiments. To optimize transformation efficiency, co-transformations were performed in two steps. Following transformation with the p15A plasmid, cells were grown the next day, then made chemically competent to transform with the pSC101 variant plasmid.

## Design of sgRNA sequences

Four sgRNAs were used in the study, targeting (i) *sfGFP*d in Fig. 2a and Fig. 3a; (ii) *ampR*d in Fig. 3e; (iii) the DNA binding domain of *lacI* in Fig. 4a and Fig. 6bc; and (iv) the P$_{tetO-lacO}$ promoter in Fig. 6bc. All sgRNA sequences were examined using Vienna RNA[94] for folding. To minimize undesired secondary structures, the sgRNA targeting *ampR*d and *lacO* contain a scaffold with a hairpin lock to stabilize the structure[95]. The other sgRNAs contain the unmodified *Streptococcus pyogenes* scaffold. The sgRNA sequences and the corresponding targeting regions are annotated in Supplementary Table 3. All four sgRNA targeting regions have also been checked in CRISPRscan[96], showing sufficient efficiencies (Supplementary Table 6).

## General protocol for flow cytometer experiments

Co-transformants of EvolvR and pSC101 variant plasmids are grown overnight on LB agar plates at 37 °C with appropriate antibiotics and 0.2% glucose (to suppress the P$_{BAD}$ promoter). Colonies are picked on day 1 for overnight growth at 37 °C in LB liquid media supplemented with antibiotics and 0.2% glucose. On day 2, each overnight culture is diluted approximately 10,000-fold into 1 mL LB with antibiotics and 2 mM arabinose, except for experiments in Fig. 6, where 200 μM arabinose is used for induction. To ensure consistent growth, the culture is first diluted approximately 30-fold to reach an OD of 0.15, and then diluted another 333-fold. Next, each 1 mL diluted culture is distributed into 8 wells of a 96-well plate, with 105 μL per well. The plate is inserted into a microplate reader (Biotek Cytation 5M) to incubate with shaking at 30 °C for 22 h (induced culture). Meanwhile, each overnight culture from day 1 is diluted 10,000-fold into LB media with antibiotics and 0.2% glucose, and grown at 37 °C for 5 h (pre-induced culture). On day 3, induced cultures are diluted 5000-fold into LB media with antibiotics and 0.2% glucose, then grown at 37 °C for 6 h or longer to reach the exponential phase. For experiments in Fig. 6, 100 ng/mL aTc is supplemented in the media during the post-induction regrowth stage to relieve the repression from residual TetR, as well as the pre-induction stage for consistent comparison. Both pre-induced and induced cultures were analyzed with a flow cytometer (Thermo Fisher Attune NxT) and the FlowCal Python package[97] with at least 100,000 events for each sample. For details on gating and identifying mutant events, see Supplementary Figs. 7, 8.

## Fluctuation assay protocol for restoring antibiotic resistance

We use the following protocol adapted from ref. 98 for the experiment in Fig. 3d to restore the function of *ampR*d. On day 0, colonies of co-transformed plasmids are picked to grow in LB media with appropriate antibiotics (kanamycin and chloramphenicol) containing 0.2% glucose at 37 °C. On day 1, the overnight culture is diluted 10,000-fold into LB media with antibiotics (kanamycin and chloramphenicol) and 2 mM arabinose. The diluted culture is transferred to a 96 well-plate by distributing it over wells, each containing 105 μL volume, and incubated with shaking in a microplate reader (Biotek Cytation 5M) at 30 °C for 22 h. On day 2, induced cultures are plated on selective agar (containing carbenicillin) and non-selective agar (containing kanamycin and chloramphenicol). For selective agar plating, 15 μL induced culture from each well is mixed with 150 μL LB containing carbenicillin, then plated on 12-well plates each containing 1.5 mL agar (with carbenicillin). For non-selective agar plating, the induced culture is diluted 1,000,000-fold, of which 500 μL is plated onto an agar Petri dish (15 mL agar). On day 3, colony forming units (CFU) are counted and used as the input to calculate the mutation probability utilizing the R package flan[99]. The experiment is repeated three times on different days.

## Sample preparation for genome and plasmid sequencing

To prepare genome samples, co-transformants of EvolvR and pSC101 variant plasmids are picked to prepare overnight cultures at 37 °C, which are subsequently induced at 30 °C with 2 mM arabinose for 22 h. Next, induced cultures are either sorted for high sfGFP mutants (for sgRNA targeting $sfGFP_d$, $lacI$, and $lacO$) using a cell sorter (BD FACSAria III), or grown on carbenicillin containing agar plates to isolate mutated colonies (for the sgRNA targeting $ampR_d$). Mutated cells are subsequently grown overnight at 37 °C and collected for genomic DNA using the QIAamp DNA kit (56304). In particular, the regrown culture is first concentrated to form a pellet and stored at −80 °C for 1 h. Then 2 μL protease K (NEB P8107S) is added to the pellet culture. The mixture is incubated at 37 °C for 2 h, followed by incubation at 56 °C for 1 h, then heat inactivated at 80 °C for 5 min. The mixture is cleaned following the steps suggested by the QIAamp DNA kit using the provided buffers and spin columns. The purified genome samples are verified on an agarose gel and shipped on dry ice for NGS genome sequencing (Novogene Singapore). Variant calling is performed on genome sequencing results, using a minimal quality score of 20 and a minimal read depth of 10 for variant detection, as recommended in ref. 100. Results are summarized in Supplementary Table 1.

To prepare plasmid samples for mutation analysis, we induce cultures for 22 h at 30 °C with 2 mM arabinose, sort the cultures for high sfGFP expressing mutants using a cell sorter (BD FACSAria III), and collect at least 1000 events for each culture. Sorted cells are regrown on agar plates to isolate individual sfGFP-expressing colonies for plasmid preparation. At least 15 colonies are collected for each sgRNA (targeting $sfGFP_d$, $lacI$, and $lacO$), and whole plasmids are subsequently analyzed using Nanopore sequencing (Plasmidsaurus).

## Statistical analysis

Centers in all figures represent mean values, except for the boxplot figures where the centers represent medians. Error bars correspond to the 95% confidence intervals throughout both the manuscript and the Supplementary Information, except for the boxplot figures. The lower and upper bounds of the box represent the first quartile (Q1) and third quartile (Q3), respectively. With IQR = Q3−Q1 denoting the interquartile range, the whiskers extend from the box to the minimum and maximum non-outlier values such that the lower and upper thresholds are defined as Q1 − 1.5IQR and Q3 + 1.5IQR, respectively.

## Software tools

For data analysis and visualization, Python (version 3.10.14) was used in the JupyterLab interface (version 4.2.2). FlowCal python library (version 1.3.0) was used to analyze flow cytometry data. R (version 4.1.2) and the flan package (version 0.9) were used for fluctuation analysis. Adobe Illustrator (version 28.5) was used for creating figures. Microsoft Word for Mac (version 16.62) was used to create the tables presented in the Supplementary Information. Overleaf (version v2) online LaTeX editor was used to prepare and compile the manuscript.

## Reporting summary

Further information on research design is available in the Nature Portfolio Reporting Summary linked to this article.

# Data availability

Source Data are provided with this paper. Sequencing data generated in this study are uploaded to NCBI SRA (PRJNA1373182). Source data are provided with this paper.

# Code availability

Data analysis and numerical simulations were performed using Python, source code is available at https://doi.org/10.5281/zenodo.17642376.

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

## Acknowledgements

We thank Howard Salis for helpful comments about the use of the RBS calculator. This work was supported by research funds from New York University Abu Dhabi. We thank the support that we received with flow cytometer experiments from the Center for Genomics and Systems Biology and the Core Technology Platform at New York University Abu Dhabi.

## Author contributions

X.L. contributed to the conception and design of the work, to the construction of the genetic circuits, and to the acquisition, analysis and interpretation of data. A.G. contributed to the conception and design of the work, and to the analysis and interpretation of data. X.L. and A.G. together created the figures and drafted the initial submission and the revisions.

## Competing interests

The authors declare no competing interests.
