## [Transparent Peer Review file · Nature Communications]

Tuning evolvability via plasmid copy number and regulatory architecture

Corresponding Author: Dr Andras Gyorgy

Version 0:

Reviewer comments:

Reviewer #2

(Remarks to the Author)

This paper aims to study the effect of plasmid copy number and various regulatory motifs affect the rate of mutation of a phenotypically silenced fluorescent protein using a two plasmid system with the CRISPR-based EvolvR mutagenesis system. The authors have developed a reasonable strategy for interrogating the phenotypic mutation rate, but there are several questions that arise from their experimental design.

First, it's not clear how the authors are tracking the genotype of individual plasmids within each cell. As the authors point out, the copy number of a plasmid can cause there to be a handful to dozens of the plasmid in a cell. As the plasmid replicates, there is intrinsic mutational variation or heterogeneity in the plasmid sequence from plasmid to plasmid within a cell (see the work of Michael Baym for example). It's not clear how the authors take this heterogeneity into account but it seems intrinsic to the goal. For example, variations in the promoter sequence or the terminator region, or even regions outside of the reporter gene that change the rate of transcription initiation could affect the measured outcomes beyond the action of just the EvolvR system. In Section 7 of the supplement, this is partially addressed by bulk NGS experiments on what seems to be the plasmids (also, what about the genome?... see my next comment below), but this is an bulk, averaged measurement of a culture and does not constitute genotyping of each plasmid. This could be clarified.

Second, it's not clear whether EvolvR finalizes its mutation in a single plasmid at the end of a gene of interest or if it continues onwards as a processive polymerase to replicate and mutate other elements of the plasmid. This probably is stated somewhere in the paper, but I could not find any discussion on the point, which seems to be a critical premise in the analysis. How do we conceptually model the EvolvR's mutational action - just confined to the gene of interest or potentially spilling over into the other elements of the plasmid? This seems to be a critical point.

Third, there will be basal mutation rates in the plasmid and the genome that confound the measured mutation rates from sfGFP expression. These effects will potentially be exacerbated by off-target binding of the EvolvR system, as many Cas-based systems have been shown to have non-specific or off-target binding to other genetic elements on the genome. None of the experiments/data in the figures of the manuscript quantify the genetic drift of the genome or attempt to control for this possibility. To assume it is entirely absent seems a bit too strong of an assumption for the goals of the study.

Lastly, it seems like having two plasmids in the same cell with a CRISPR system on one would impose a potential metabolic burden on the cell, especially after activation with a metabolically relevant compound such as arabinose. I did not see strong evidence of the orthogonality of these two-plasmid systems to the fitness and metabolism of the cell. Some basic control experiments measuring fitness or impact would improve the rigor of the paper's argument.

In general, it seems like a thoughtful series of arguments in the introduction or perhaps the results section are missing to justify the rigor of the experimental approach presented. The results are interesting and merit publication, but there are some minor yet basic, conceptual revisions here that need to be addressed prior to publication.

Minor comments:

It is not clear if the content in Figures 3 and 4 are wholly based off of stochastic simulation analysis or experiment. The delineation between experiment and simulation could be clearer in Figure captions.

Reviewer #3

(Remarks to the Author)

This manuscript tackles a novel question in synthetic biology on the impact of plasmid mutations on the behavior of circuits and regulatory motifs, and how this relationship is itself impacted by circuit topology and plasmid copy number. This study has the potential to help significantly improve the robustness of synthetic circuits to mutations, as well as improve performance in directed evolution applications. I find that the authors could better introduce the novelty of their study in the introduction and its potential impact. The authors could also better introduce the motivation behind each of the topologies they test throughout the study. The manuscript is otherwise well written and the results are presented clearly.

My main concern is that the authors use a statistics-based method to identify the phenotypic mutation rate in their cultures (Supplementary Fig. 2). This method seems sound and it is at the center of most results in this manuscript, but it needs to be described more thoroughly both in the Main text and in the Supplementary text. I assume they are using the pre-induced cultures that are grown in parallel, mentioned in the Methods section (l. 360-362), as reference for their t-tests but it is not explicitly said anywhere in the text. However the pre-induced cultures surprisingly do not follow the same protocol as the induced cells, as they grow under different temperatures and durations, which might bias the t-tests. The flow cytometry data collected for these pre-induced reference populations should also be shown in Supplementary Fig. 2 and 6-12.

Minor comments:

Line 84: "After their duplication (STEP 2 in Fig. 1a), plasmids from..." the duplication step appears to be deterministic, where the number of plasmids exactly doubles, and all plasmids are duplicated. This is not consistent with actual plasmid replication mechanisms, did the authors consider making the duplication probabilistic in their model?

On Figures 2-6 the authors should describe what the error bars represent and give the number of replicates (8?). For all phenotypic mutation plots, the authors should also consider testing for statistical significance between, at least between PCN=3 and PCN=39/79.

Figure 3a & 3e: "p15A harboring EvolvR and sfGFP" -> sfGFP is not on that plasmid.

Lines 237-244: This part is speculative and the same issue should also happen in Fig. 5b. The authors should either test the binding affinity of their mutated promoters, or rephrase this section.

In Supplementary Fig. 5, the authors should specify which plasmid copy number variant they used. This Figure should also be referenced earlier in the Main text to support the validity of their approach.

Version 1:

Reviewer comments:

Reviewer #2

(Remarks to the Author)

I appreciate the authors thoughtful responses, their additional experiments, and their dutiful follow-up on the questions I raised regarding intracellular plasmid variability. The authors have addressed the majority of my concerns. My primary concern is still regarding the framing of this paper. Figure 1 introduces the theme of this paper as intracellular plasmid variability and presents beautiful simulations to quantify the statistics of accumulation of mutation. As I see it (and I may have missed something), no other figure in the paper continues on this vein, but rather speaks towards general phenotypic outcomes of mutation using the EVOLVR system cleverly developed by the authors.

The supplementary figure 5 shows the outcomes of whole plasmid sequencing experiments on individual colonies, which are presumed to be isogenic variants in the traditional sense. However, the issue of intracellular plasmid variability as introduced in Figure 1 is still unaddressed, in this regard. A single cell can have multiple mutants of a plasmid and whole plasmid sequencing, NGS methods do not quantify this intrinsic genetic variation but rather present a measurement of a bulk sample of hundreds of thousands of cells.

I agree with the authors that development of methodologies to measure such intrinsic variability would be outside the scope of this work, but I think Figure 1 is quite confusing, in that it sets the expectation that the focus of this work will be on that theme. The subsequent figures are fine and tell a strong story, but the framing of the paper just needs some adjustment. No additional experiments are required, as I can tell.

One minor comment: In Supplementary Figure 5 (and elsewhere throughout the paper), standard error is used to plot the error bar. This is misleading, as the standard error represents the standard deviation normalized by an additional sampling factor n — (\sqrt{n}) to be precise). I recommend the authors use 95% confidence intervals instead, as standard deviation is often too generous of a measure while standard error is too conservative. Using a confidence interval is generally statistically more rigorous, as well.

Reviewer #3

(Remarks to the Author)

The authors have thoroughly and convincingly addressed my comments and concerns. I recommend this study for publication in Nature Communications.

Response to Referees

We thank the Editor and the Reviewers for their constructive feedback that enabled us to enhance the rigor and clarity of our manuscript. We undertook substantial experimental, computational, and editorial revisions to address the helpful comments that we have received (highlighted in blue). In particular:

1. **Expanded experimental validation:** We performed new whole-plasmid sequencing of individual mutant colonies to address plasmid-level heterogeneity, quantified background mutation rates using genome sequencing, and measured metabolic burden via growth rate assays. These additions directly addressed concerns about mutation specificity, off-target effects, and potential confounding factors.
2. **Clarified EvolvR mutational scope:** We explicitly defined EvolvR's editing window, provided experimental evidence on mutation confinement, and implemented a TetR-based repression module with reduced induction levels to mitigate off-target activity.
3. **Enhanced computational analyses:** We integrated stochastic plasmid replication into the model, modeled plasmid genotype distributions within single cells, and provided detailed justifications for parameter choices. These changes improved biological realism without altering key conclusions.
4. **Strengthened methodological transparency:** We expanded descriptions of the phenotypic mutation rate estimation, included all pre-induction flow cytometry controls, clarified statistical testing procedures, and updated figure captions to explicitly distinguish between experimental and simulated datasets.
5. **Improved contextual framing and rationale:** The Introduction and Discussion were expanded to better articulate the novelty, impact, and design motivation for specific circuit topologies (incoherent feedforward loop and negative autoregulation), in addition to linking our results to broader synthetic biology and directed evolution principles.
6. **Refined data presentation:** Figures were reorganized to separate simulations from experiments, replicate numbers and error bar definitions were added, and statistical comparisons across conditions were included.

Collectively, we believe that these changes address all Reviewer concerns, reinforce the scientific rigor of our approach, and clarify the broader significance of the findings. In addition to the revised manuscript, we also uploaded a copy indicating all the changes (generated using the `latexdiff` package: added text is blue, discarded text is red).

Reviewer #2

This paper aims to study the effect of plasmid copy number and various regulatory motifs affect the rate of mutation of a phenotypically silenced fluorescent protein using a two plasmid system with the CRISPR-based EvolvR mutagenesis system. The authors have developed a reasonable strategy for interrogating the phenotypic mutation rate, but there are several questions that arise from their experimental design.

We thank the Reviewer for recognizing the merit of our strategy and for providing us with constructive and helpful comments regarding our experimental design. In the revised manuscript, we have substantially expanded both experimental and computational analyses to address these concerns. Specifically, we now:

1. **Address plasmid heterogeneity:** We performed additional simulations (Supplementary Fig. 2) to model the distribution of genotypes across plasmid copies within single cells, and conducted whole-plasmid sequencing from isolated mutant colonies replacing bulk NGS measurements (Supplementary Figs. 5–6).
2. **Clarify EvolvR's mutational scope:** We explicitly state that EvolvR edits within ~ 60 nt of the Cas9 nick site, and present sequencing data showing that mutations are predominantly on-target, with only occasional spillover into adjacent plasmid elements (Supplementary Figs. 5–6).
3. **Quantify background mutation rates:** Using literature-derived rates and genome sequencing of induced samples, we show that EvolvR activity causes virtually no genomic mutations under our experimental conditions (Supplementary Section 2.3, Supplementary Table 1).
4. **Assess metabolic burden:** We now include growth rate measurements (Supplementary Fig. 4) showing that arabinose induction in the absence of EvolvR does not affect growth, whereas EvolvR expression does impose a measurable burden.

Finally, we reorganized the Results and Discussion sections to improve the logical flow and rigor, and updated figure captions to clearly distinguish between experimental and simulation data. We believe these additions resolve the conceptual uncertainties identified and strengthen the foundation of our study.

First, it's not clear how the authors are tracking the genotype of individual plasmids within each cell. As the authors point out, the copy number of a plasmid can cause there to be a handful to dozens of the plasmid in a cell. As the plasmid replicates, there is intrinsic mutational variation or heterogeneity in the plasmid sequence from plasmid to plasmid within a cell (see the work of Michael Baym for example). Its not clear how the authors take this heterogeneity into account but it seems intrinsic to the goal. For example, variations in the promoter sequence or the terminator region, or even regions outside of the reporter gene that change the rate of transcription initiation could affect the measured outcomes beyond the action of just the EvolvR system. In Section 7 of the supplement, this is partially addressed by bulk NGS experiments on what seems to be the plasmids (also, what about the genome?.. see my next comment below), but this is an bulk, averaged measurement of a culture and does not constitute genotyping of each plasmid. This could be clarified.

To address the issue of intrinsic plasmid heterogeneity, we performed additional experiments as well as computational analysis. While assessing the performance of the EvolvR system does not constitute a central theme of our manuscript, this additional investigation complements the extensive work that was undertaken during the development of this *in vivo* mutagenesis platform by Halperin *et al.* in their article, cited as (47) in our manuscript.

From a computational perspective (Supplementary Section 1), to estimate the distribution of genetic variations at the plasmid level, we first note that EvolvR mutates the target at a rate of $2.5 \cdot 10^{-6}$ per nucleotide per generation, suggesting that mutations are sparse across generations. This aligns well with the scenario underpinning fluctuation analysis, in which a few early-generation mutations expand into descendants. Consequently, many mutant plasmids are expected to share similar genotypes from a few mutated ancestors that arise early on. To investigate the issue of intrinsic plasmid heterogeneity, we categorized mutated plasmids based on the generation in which each mutation first occurred. We assume that each plasmid mutates at most once, as the EvolvR-associated sgRNA loses binding affinity to a mutated target. Thus, each mutated plasmid is assigned to a unique generation. The distribution of these generations (that eventually gives rise to the distinct mutations we observe in the final simulated generation) is presented in Supplementary Fig. 2. As we now state in the manuscript (lines 101–103), these data reveal that “*while mutations that arise in early generations tend to dominate the mutant population for low PCN, increasing gene dosage contributes to greater genotypic diversity with mutations emerging across multiple generations (Supplementary Fig. 2).*”

From an experimental perspective (Supplementary Section 2.2), since next-generation sequencing (NGS) measurements characterize mutation genotypes that emerge only in bulk at the population level, we isolated individual colonies exhibiting mutant phenotypes and prepared the plasmids for sequencing. In particular, cells were grown after arabinose induction, and following regrowth, the ones showing GFP expression were sorted and regrown on agar plates (see Methods). Thus, plasmids in the resulting individual colonies are expected to be genetically homogeneous, hence sequencing them provides fine-grained genetic information that is not available via bulk NGS. As it is now included in the manuscript (lines 109–110): “*EvolvR is designed to edit within a window extending to around 60 nucleotides in the 3' direction from the Cas9 nick site (47),*” together with additional details in Supplementary Section 2.2 to acknowledge that mutations can potentially occur beyond this range with decreasing efficiency and that the size of this window depends on the processivity of the error-prone DNA polymerase in EvolvR (PolI3M), and can be further modulated via the insertion of a thioredoxin-binding domain (TBD) into the polymerase (PolI3M-TBD). Mutations that occur within the intended region are considered on-target, otherwise they are off-target. On-target ratios are calculated by dividing the on-target mutation counts with respect to the total number of mutations. As we now state in the manuscript (lines 120–122): “*EvolvR activity is largely confined to the target editing window, correcting the premature stop codon and restoring function resulting in the dominance of a small subset of genotypes (Supplementary Fig. 5b).*”

Finally, we surveyed the relevant literature as suggested by the Reviewer and acknowledge in the Discussion (lines 299–302) that “*the role and impact of intrinsic plasmid heterogeneity can be further studied using advanced experimental techniques such as induced plasmid monomerization (90) or via Cas9-based lineage recording (91) to genetically encode plasmid replication events.*” This line of inquiry, however, lies outside of the scope of our manuscript.

Second, it's not clear whether EvolvR finalizes its mutation in a single plasmid at the end of a gene of interest or if it continues onwards as a processive polymerase to replicate and mutate other elements of the plasmid. This probably is stated somewhere in the paper, but I could not find any discussion on the point, which seems to be a critical premise in the analysis. How do we conceptually model the EvolvR's mutational action - just confined to the gene of interest or potentially spilling over into the other elements of the plasmid? This seems to be a critical point.

To clarify the expected behavior of the mutagenesis platform, we now explicitly state in the manuscript (lines 109–110) that “*EvolvR is designed to edit within a window extending to around 60 nucleotides in the 3' direction from the Cas9 nick site (47),*” alongside with further details in Supplementary Section 2.2.

Additionally, to examine whether mutations are confined to this window, we sequenced whole plasmid DNAs from individual colonies that show mutant phenotypes (as detailed in the previous comment). For the sgRNA targeting GFP_d (Fig. 2a and Fig. 3a) we now include that (lines 120–122) “*EvolvR activity is largely confined to the target editing window, correcting the premature stop codon and restoring function resulting in the dominance of a small subset of genotypes (Supplementary Fig. 5b).*” Complementing this, we also acknowledge (Fig. 4 caption) that “*While off-target mutations become more prevalent in case of LacI than when targeting GFP_d, the majority of these unintended mutations are still restricted to the lacI gene, giving rise to the same mutation phenotype (Supplementary Fig. 5).*” Finally, we observe decreased EvolvR specificity for the sgRNA recognizing the lacO binding sites (Supplementary Fig. 6), which is not surprising, and is now stated in the manuscript (lines 233–237): “*as Cas9-based systems are known to suffer from off-target effects, especially when binding sites become saturated due to the high relative prevalence of Cas9 with respect to its target (65–67). This likely occurs when the sgRNA targets the promoter, as LacI binding to lacO blocks EvolvR from accessing the promoter (68), diverting EvolvR to unintended sites and resulting in off-target mutations (Supplementary Fig. 6).*” To address this issue (lines 231–232), “*we integrated a TetR-based repression module into the two-plasmid system to repress both LacI and GFP expression during EvolvR activation*” for the constructs in Fig. 6. Additionally, as we now include in the caption of Fig. 6: “*Arabinose induction is reduced from 2 mM to 200 μM to decrease the activity of EvolvR and the chance of its binding to off-target sites. Subsequent to arabinose induction, anhydrotetracycline (aTc) at a final concentration of 100 ng/mL is supplemented to relieve the effects of residual TetR.*” As noted in the caption of Fig. 6c, “*On-target mutations are found at a few specific positions that are functionally important, disrupting the lacO binding sites, resulting in the dominance of a small subset of genotypes (Supplementary Fig. 6a),*” alongside with the acknowledgment that “*even with the inclusion of the TetR repression module and the reduced EvolvR expression, off-target mutations persist for pSCI101 variants with PCN below 9.*” Therefore, the corresponding data are omitted in Fig. 6c, and instead included in Supplementary Figs. 17–18.

The data in Supplementary Figs. 5–6 alongside with the extensive characterization during the original development of EvolvR illustrate (as we now state in the Discussion, lines 286–292) that “*EvolvR activity is mostly confined to the gene of interest (though not necessarily to the editing window), causing virtually no genomic mutations, only occasionally spilling over into other elements of the plasmid. Our data also highlight that besides introducing an additional control layer (such as our TetR-based repression module) or reducing EvolvR expression, increasing the abundance of the target site via PCN control can offer an effective alternative strategy to combat this undesirable characteristic of Cas9-based tools.*”

Third, there will be basal mutation rates in the plasmid and the genome that confound the measured mutation rates from sfGFP expression. These effects will potentially be exacerbated by off-target binding of the EvolvR system, as many Cas-based systems have been shown to have non-specific or off-target binding to other genetic elements on the genome. None of the experiments/data in the figures of the manuscript quantify the genetic drift of the genome or attempt to control for this possibility. To assume it is entirely absent seems a bit too strong of an assumption for the goals of the study.

Complementing the work outlined in the previous two comments addressing plasmid heterogeneity and EvolvR's targeting accuracy around the editing window, we included additional computational analysis and experimental verification to estimate the basal mutation rates in the plasmid and the genome that could confound the measured mutation rates.

From a computational perspective, considering previously reported mutation rates, we estimate less than a single nucleotide mutation in the genome throughout our experiments. In particular, as we detail in Supplementary Section 2.3, *E. coli* has a basal spontaneous mutation rate of approximately 10^{-10} mutations per nucleotide per generation. The Cas9 variant in EvolvR (enCas9-PolI3M-TBD) is engineered so that it has reduced non-specific DNA binding, resulting in on-target and global mutation rates of approximately 10^{-5} and 10^{-8} mutations per nucleotide per generation, respectively. We utilized a P_{BAD} -inducible system to tightly regulate EvolvR and sgRNA expression, hence the approximate background mutation rate can be estimated as follows. Prior to induction, cultures are grown for around 24 hours (i.e., 32 generations at 37°C) on agar (12 hours) and liquid media (12 hours) with 0.2% glucose (Fig. 2b) to suppress basal P_{BAD} expression through catabolite repression, ensuring a low spontaneous mutation rate during this stage. Subsequently, we induce EvolvR expression using arabinose for 22 hours at 30°C, i.e., for approximately 15 generations. Considering the typical *E. coli* genome size, this is expected to generate approximately $5 \cdot 10^6 \cdot (10^{-10} \cdot 32 + 10^{-8} \cdot 15) = 0.766$ mutations, i.e., less than a single mutation throughout the experiment (and even fewer for the plasmids considering their maximal size and PCN: $13 \cdot 12 \cdot 10^3 \cdot (10^{-10} \cdot 32 + 10^{-8} \cdot 15) = 0.024$ for p15A with 13 copies and $100 \cdot 8 \cdot 10^3 \cdot (10^{-10} \cdot 32 + 10^{-8} \cdot 15) = 0.123$ for pSC101 with no more than 100 copies).

To validate this estimate, we collected mutated samples after arabinose induction and performed genome sequencing to identify potential mutations (targeting GFP_d , $ampR_d$, $lacI$, and $lacO$). As the sgRNA targeting the $lacI$ gene harbored on the plasmid also recognizes the one in the DH5 α genome, we generated the DH5 $\alpha\Delta lacI::ampR$ strain by knocking out the genomic $lacI$ gene. Following this, we performed variant calling on the genome sequences obtained using next generation sequencing (see Methods). Among all tested samples, only a single mutation was detected in the genome (Supplementary Table 1), when the sgRNA targets GFP_d , yielding a single nucleotide polymorphism (A to T) in the PqqL gene encoding for a zinc protease (turning the 122nd amino acid from Val to Glu; no nearby sequence resembles the sgRNA target, but several NGG motifs are present). These data confirm the low background mutation level suggested by our computational estimate.

This is now explicitly stated in the Discussion (lines 286–288): “*EvolvR activity is mostly confined to the gene of interest (though not necessarily to the editing window), causing virtually no genomic mutations, only occasionally spilling over into other elements of the plasmid.*”

Lastly, it seems like having two plasmids in the same cell with a CRISPR system on one would impose a potential metabolic burden on the cell, especially after activation with a metabolically relevant compound such as arabinose. I did not see strong evidence of the orthogonality of these two-plasmid systems to the fitness and metabolism of the cell. Some basic control experiments measuring fitness or impact would improve the rigor of the paper's argument.

To address this shortcoming, we performed additional microplate reader experiments to quantify the impact of arabinose induction, as well as the metabolic burden that each of the plasmids impose on the host. Growth rate data in Supplementary Fig. 4 highlight that (i) induction via arabinose in the absence of EvolvR on the p15A plasmid or harboring the target plasmid expressing GFP_d from a pSC101 plasmid do not impact cell growth; whereas (ii) the expression of EvolvR from a p15A plasmid does. This is now also acknowledged in the caption of Fig. 2b.

In general, it seems like a thoughtful series of arguments in the introduction or perhaps the results section are missing to justify the rigor of the experimental approach presented. The results are interesting and merit publication, but there are some minor yet basic, conceptual revisions here that need to be addressed prior to publication.

We appreciate the Reviewer's constructive feedback and helpful recommendations. As outlined above, we have taken several steps to justify the rigor of our experimental approach: additional data collected during the revision process have been integrated into the manuscript and Supplementary Information to reinforce key claims, which range from growth rate measurements to sequencing analysis. These additions, complementing the changes that we made to the manuscript itself to improve the logical progression from hypothesis to experimental design, and from data to conclusions, render our findings more transparent to the reader. We believe that together these revisions address the Reviewer's concern and strengthen the overall rigor of our study.

Minor comments: It is not clear if the content in Figures 3 and 4 are wholly based off of stochastic simulation analysis or experiment. The delineation between experiment and simulation could be clearer in Figure captions.

To improve clarity and the delineation between experiments and simulations, each figure caption now explicitly states whether the results are obtained in simulations or experiments, in addition to the complete reorganization of Figs. 5–6 to separate the two types of data.

Reviewer #3

This manuscript tackles a novel question in synthetic biology on the impact of plasmid mutations on the behavior of circuits and regulatory motifs, and how this relationship is itself impacted by circuit topology and plasmid copy number. This study has the potential to help significantly improve the robustness of synthetic circuits to mutations, as well as improve performance in directed evolution applications.

We thank the Reviewer for their constructive and encouraging feedback, which recognized the novelty and potential impact of our study on understanding how plasmid copy number and regulatory architecture influence mutation dynamics in synthetic circuits. These insightful comments have significantly strengthened the manuscript and helped us present our findings with greater clarity and rigor. In particular, as we detail in our point-by-point response, we made the following changes to address the helpful points raised by the Reviewer:

1. **Expanded Introduction and Discussion:** We provided further context to articulate the novelty and potential impact of the study, emphasizing how our results can inform synthetic biology design and directed evolution strategies, as well as to better connect the experimental and computational results to broader design principles for building mutation-resilient circuits and tuning evolvability.
2. **Motivation for circuit topologies:** We included the rationale for integrating the incoherent feed-forward loop and negative autoregulation motifs into our constructs, explaining their relevance for modulating gene expression, evolutionary adaptation, and mitigating metabolic burden.
3. **Clarified phenotypic mutation rate methodology:** We included detailed explanations describing the statistical framework, controls, and data sources for fluctuation analysis. We also incorporated pre-induction flow cytometry data for completeness and transparency.
4. **Additional experimental and computational validation:** We conducted further sequencing experiments to characterize the off-target impact of EvolvR, integrated an additional control module to mitigate these effects, and incorporated stochastic plasmid replication into the simulations.
5. **Improved data presentation:** We updated figure captions to clearly distinguish simulation from experimental datasets, added the number of replicates and error bar definitions, performed pairwise statistical tests between PCN conditions, and explicitly stated the PCN throughout both the manuscript and the Supplementary Information.

We sincerely thank the Reviewer for their thoughtful and detailed feedback, which has directly contributed to improving both the scientific depth and clarity of our manuscript. Their suggestions prompted us to expand our contextual framing, refine our experimental and computational methodologies, and improve the transparency of our data presentation. Furthermore, the encouragement to provide deeper motivation for our design choices and to validate our findings through additional experiments has strengthened the robustness of our conclusions. The revisions inspired by these comments not only address the specific points raised but also broaden the scope and potential impact of our study within the fields of synthetic biology and directed evolution.

I find that the authors could better introduce the novelty of their study in the introduction and its potential impact. The authors could also better introduce the motivation behind each of the topologies they test throughout the study. The manuscript is otherwise well written and the results are presented clearly.

To address this shortcoming, we revised both the Introduction and the Discussion to better introduce the novelty of our study and its potential impact.

We first incorporated the following in the Introduction (lines 53–56): “*Harnessing EvolvR, we implemented a collection of engineered circuits to study the emergence of plasmid-borne mutations. We show that the interplay between gene dosage and regulatory motifs shapes both genotypic and phenotypic heterogeneity, highlighting the value of integrating synthetic gene circuits and in vivo mutagenesis to study evolutionary dynamics.*” Additionally, we also included in the Discussion (lines 251–256) that “*it is imperative to understand how circuit-level features link genetic mutations to the spread of mutants in the presence of inevitable evolutionary forces to uncover the organizing principles of living cells, and to devise design guidelines that ensure the predictable behavior of engineered biocircuits. This is especially crucial when evolving genetic circuits at an intermediate scale between focusing on optimizing single genetic components and facilitating genome-wide adaptations in serially propagated cell populations (76).*”

These additions clarify the novelty of our work and better contextualize and highlight the potential impact of our results, complementing the closing paragraph of the Discussion (lines 305–312): “*Our work provides explicit design guidelines by revealing how regulatory architecture and gene dosage via PCN together shape the emergence of mutants within a population, despite the simplicity of our computational model (e.g., mutations can be neutral or even increase protein stability and enhance binding (92, 93)) and the imperfect nature of the mutation platform (e.g., off-target effects of EvolvR (47)). Therefore, the results presented here shed light on evolutionary driving forces, and can be leveraged to guide the design of evolutionarily stable synthetic gene circuits or to dynamically program genetic diversification by leveraging increased evolvability (88, 89), for instance, via flexible PCN control (81, 94).*”

Finally, we provide further motivation and detail how the the regulatory motifs used in our study (i.e., IFFL and negative autoregulation) impact circuit performance and evolvability throughout the manuscript:

- lines 191–192: “*Autoregulated motifs play a fundamental role in modulating gene expression and shaping evolutionary adaptation (10, 54–59).*”
- lines 263–267: “*Integrating an additional control layer (like the IFFL-based regulatory module or negative autoregulation) can relax such bioenergetic constraints (57), allowing the circuits to persist on high copy plasmids. This in turn can directly impact their evolvability among other benefits (58, 59), likely contributing to the ubiquitous nature of these motifs in transcription networks (82, 83).*”
- lines 276–280: “*Additionally, inclusion of a control layer can modulate the mapping from genotypic mutations to phenotypic mutation rate depending on the detection threshold, which could be crucial when establishing genetic dominance (85). That is, by regulating gene expression, we can adjust the visibility and significance of inevitable mutations, effectively altering the dominance of genetic traits (85).*”

My main concern is that the authors use a statistics-based method to identify the phenotypic mutation rate in their cultures (Supplementary Fig. 2). This method seems sound and it is at the center of most results in this manuscript, but it needs to be described more thoroughly both in the Main text and in the Supplementary text. I assume they are using the pre-induced cultures that are grown in parallel, mentioned in the Methods section (l. 360–362), as reference for their t-tests but it is not explicitly said anywhere in the text. However the pre-induced cultures surprisingly do not follow the same protocol as the induced cells, as they grow under different temperatures and durations, which might bias the t-tests. The flow cytometry data collected for these pre-induced reference populations should also be shown in Supplementary Fig. 2 and 6–12.

We thank the reviewer for pointing out this shortcoming. To clarify these sources of confusion, we have made the following three changes throughout the manuscript and the Supplementary Information.

First, we provide all pre-induction flow cytometer data alongside post-induction measurements in Supplementary Figs. 19–25. Growth of both cultures involve dilution followed by regrowth at 37°C prior to FACS analysis. The major difference between pre- and post-induction workflows is that the latter requires longer regrowth, caused mainly because EvolvR proteins accumulate during induction and continue to exist in cells, and the corresponding metabolic burden negatively impacts cell growth (Supplementary Fig. 4). To mitigate this stress, we adopted 30°C during the induction stage instead of 37°C (as indicated in Fig. 2b and explicitly stated in the figure caption). Additionally, the regrowth time may also vary depending on several other factors, including final cell density after induction, manual dilution, batch media quality and the uneven temperature distribution in shaker incubators. We therefore allow a flexible regrowth window that spans approximately 6 hours, ranging from 5 to 7 hours. This approach ensures that cultures reach comparable growth states in exponential phase before subsequent FACS analysis for mutant detection (which is also included in the caption of Fig. 2). The protocol is included in Fig. 2b for easy comparison of the two phases, and it is explicitly stated in the figure caption that the mutant subpopulation is identified using flow cytometer data comparing the pre- and post-induced samples.

Second, Supplementary Fig. 8 has also been refined for enhanced clarity, including the following in the caption: *“Cultures were screened prior to arabinose induction as a control to exclude spontaneously mutated individuals and to provide a reference for mutant detection.”*

Third, we revised the description of how mutants are detected in Supplementary Section 3. Key modifications are the inclusion of the details that underpin the statistics-based method for identifying mutant cells, as well as further steps on the calculation of the phenotypic mutation rate. In particular, we leveraged the flow cytometer data to quantify mutant cells, which were then used to calculate phenotypic mutation rate using fluctuation analysis. The mutation rate represents the probability of mutation per generation, estimated from the distribution of the observed mutants at the final time point.

Minor comments:

Line 84: “After their duplication (STEP 2 in Fig. 1a), plasmids from. . .” the duplication step appears to be deterministic, where the number of plasmids exactly doubles, and all plasmids are duplicated. This is not consistent with actual plasmid replication mechanisms, did the authors consider making the duplication probabilistic in their model?

To better capture the biological variability of plasmid replication and enhance the general applicability of the computational framework that underpin our simulation data, we extended the model to include the stochastic nature of plasmid duplication rather than assuming deterministic doubling as follows (lines 76–94, relevant portion appears in bold here): *“Our computational framework (Fig. 2a) has discrete and non-overlapping generations with a constant population size N , and each member is replaced in every generation as follows. Let x_m and $x_{wt} = P - x_m$ denote the number of mutated and wild-type plasmids in a given cell, respectively, where P is the PCN. Similarly, let N_m and $N_{wt} = N - N_m$ denote the number of mutant (with at least one mutated plasmid) and wild-type bacteria (those without any mutated variants), respectively. We generate the number of new mutations Δx within each cell as follows: each wild-type plasmid becomes mutated with probability μ , modeled as a Poisson process. Next, the number of wild-type and mutated plasmids for each cell are updated (STEP 1 in Fig. 1a) as $x'_{wt} = x_{wt} - \Delta x$ and $x'_m = x_m + \Delta x$, respectively. **Subsequently, the number of new plasmids P' after replication is generated according to a second Poisson distribution, with its mean equal to the average PCN (see Methods), and this pool is split according to a binomial distribution with parameters P' and x'_m/x'_{wt} , that is, newly replicated plasmids are randomly assigned to be either wild-type or mutated according to the respective proportions within the cell (STEP 2 in Fig. 1a). Following this, both daughter cells receive either $(P + P')/2$ plasmids if $P + P'$ is even, otherwise one of them ends up with one more than the other to ensure that there is no segregational loss, where a hypergeometric distribution (50) determines how mutated and wild-type plasmids are distributed (STEP 3 in Fig. 1a). Finally, half of the population is randomly selected to initialize the next generation (STEP 4 in Fig. 1a), while the rest is discarded to keep the population size constant across generations.”***

Additionally, to justify the choice of the Poisson distribution governing the stochastic nature of plasmid replication, we have included the following in the Methods (under “Stochastic plasmid replication”): *“At each generation, new plasmids are produced in each cell following a Poisson distribution with the mean equal to the PCN of the corresponding pSC101 variant, as has been previously developed for R1 plasmids (96) since both systems use RepA-mediated plasmid replication to control PCN. While the specifics of the replication mechanisms differ in R1 and pSC101 plasmids (co-regulation of RepA synthesis via CopA and CopB in the former, and negative autoregulation of RepA expression in the latter), they are both underpinned by the same feature: replication commences when sufficient RepA initiator protein binds to the iterons within the origin of replication, and the process is regulated via the repression of RepA to stabilize PCN.”*

Finally, we updated Fig. 1a to reflect the above changes in the mathematical model, as well as the simulation data throughout both the manuscript and the Supplementary Information. Importantly, while incorporating stochastic plasmid replication into our computational framework captures an important aspect of the underlying cellular processes, it does not impact the overall trends and our conclusions.

On Figures 2–6 the authors should describe what the error bars represent and give the number of replicates (8?). For all phenotypic mutation plots, the authors should also consider testing for statistical significance between, at least between PCN=3 and PCN=39/79.

To address these shortcomings, we made the following changes. First, to clarify how the phenotypic mutation rate is calculated and the meaning of the error bars, we updated Fig. 2c. In particular, we in-

clude data on the detected mutant ratio for each PCN, and state in the figure caption that “*The mutant ratio is obtained in flow cytometer experiments (Supplementary Fig. 8) using 8 replicates, which is then leveraged using fluctuation analysis (52–53) to derive the phenotypic mutation rate (error bars represent the standard deviation of the estimated mutation probability).*” Additionally, in Supplementary Section 3 we include a detailed explanation of applying fluctuation analysis to estimate the phenotypic mutation rate from mutant detection experiments using flow cytometer data. Finally, we assessed the statistical significance of the observed variation in the phenotypic mutation rate across PCN conditions using pairwise t-tests. The resulting p-values are presented as heatmaps in Supplementary Fig. 11–18, which are referenced in the corresponding figure captions of the manuscript.

Figure 3a & 3e: “p15A harboring EvolvR and sfGFP” → sfGFP is not on that plasmid.

We thank the reviewer for pointing out this error, the figure has been updated accordingly.

Lines 237–244: This part is speculative and the same issue should also happen in Fig. 5b. The authors should either test the binding affinity of their mutated promoters, or rephrase this section.

Motivated by this helpful comment, we performed additional experiments. We found that while mutations are mostly confined to the editing window when targeting GFP_d or lacI (Supplementary Fig. 5), the sgRNA targeting the lacO promoter tends to cause off-target mutations including in the lacI gene, resulting in phenotypes similar to lacO mutants (Supplementary Fig. 6).

As we detail in Supplementary Section 2.2: “*Cas9-based systems are known to suffer from off-target effects, especially when binding sites become saturated due to the high relative prevalence of Cas9 with respect to its target (4). Therefore, it is not surprising that we observe decreased EvolvR specificity for the sgRNA recognizing the lacO binding sites, especially when the PCN harboring the target sequence is low (Supplementary Fig. 6a). To understand why this happens, unlike when targeting GFP_d or lacI, note that LacI binding to lacO blocks EvolvR from accessing the promoter (5), diverting it to unintended sites and resulting in off-target mutations, for instance, in the lacI gene (Supplementary Fig. 6bc). Inducers such as IPTG can relieve LacI from binding to lacO, however, this leads to constitutive GFP expression, which would accumulate over generations and interfere with downstream mutant detection using flow cytometry. Thus, we instead integrated a TetR repression module into the two-plasmid system to repress both LacI and GFP expression during EvolvR activation (Fig. 6a). By blocking LacI production this way, the lacO binding sites become accessible for EvolvR to initiate on-target mutations. Additionally, we lowered the expression level of EvolvR by using 200 μM arabinose instead of 2 mM to reduce the chance of EvolvR binding to off-target sites. Just as before, on-target mutations are found at a few specific positions that are functionally important, in this case, to disrupt the lacO binding sites, resulting in the dominance of a small subset of genotypes (Supplementary Fig. 6a).*”

This is now also briefly included in the manuscript (lines 231–240): “*Additionally, we integrated a TetR-based repression module into the two-plasmid system to repress both LacI and GFP expression during EvolvR activation. We included this additional layer of control as Cas9-based systems are known to suffer from off-target effects, especially when binding sites become saturated due to the high relative prevalence of Cas9 with respect to its target (65–67). This likely occurs when the sgRNA targets the promoter, as LacI binding to lacO blocks EvolvR from accessing the promoter (68), diverting EvolvR to*

unintended sites and resulting in off-target mutations (Supplementary Fig. 6). By modifying the promoters of the circuits in Fig. 5 to include additional repression via TetR, the transcription of GFP and LacI are repressed upon arabinose induction both when LacI is expressed constitutively (Fig. 6a) and when it is self-repressed (Fig. 6b)." Additionally, in the caption of Fig. 6 we also state that *"Arabinose induction is reduced from 2 mM to 200 μ M to decrease the activity of EvolvR and the chance of its binding to off-target sites."*

As expected based on the simulation data presented in Fig. 5, the highlighted discrepancy between constitutive and self-repressed LacI expression (previously Figs. 5b and 6b, now Figs. 6b and 6c) is now resolved when relying on the redesigned genetic circuits in Fig. 6a. In particular, as we state in the manuscript (lines 240–243): *"Data in Fig. 6bc confirm the theoretical predictions underpinned by the simulations in Fig. 5: mutations in the coding region are masked at the phenotypic level, whereas those in the regulatory region become more prevalent as PCN increases."*

In Supplementary Fig. 5, the authors should specify which plasmid copy number variant they used. This Figure should also be referenced earlier in the Main text to support the validity of their approach.

In the revised manuscript, we now state the copy number of each plasmid variant when presenting the sequencing data that we leveraged to characterize plasmid heterogeneity and to quantify the background mutation rate, following the helpful recommendations of Reviewer #2.

First, we isolated individual colonies exhibiting mutant phenotypes and prepared the plasmids for sequencing due to the limited resolution of NGS measurements characterizing mutation genotypes that emerge only in bulk at the population level. Accordingly, Supplementary Fig. 5 is now replaced by Supplementary Figs. 5–6. To obtain these data, we sorted the arabinose induced cultures, regrew them on agar plates, and isolated the colonies showing mutant phenotypes. Plasmids were then collected from individual colonies for whole plasmid sequencing (see Methods). The captions of Supplementary Figs. 5–6 explicitly state the PCN of the variants we used.

Second, to validate our computational estimate of the background mutation rate (Supplementary Section 2.3), we collected mutated samples after arabinose induction for genome sequencing to identify potential mutations (targeting GFP_d, ampR_d, lacI, and lacO). Following this, we performed variant calling on the genome sequences obtained using NGS (see Methods). The results of the analysis are summarized in Supplementary Table 1, alongside with the PCN of the variants we used.

Taken together, the sequencing data support the validity of our approach, which is now highlighted earlier when introducing the *in vivo* mutagenesis platform (lines 120–122): *"EvolvR activity is largely confined to the target editing window, correcting the premature stop codon and restoring function resulting in the dominance of a small subset of genotypes (Supplementary Fig. 5b),"* as well as emphasized again in the Discussion (lines 286–288): *"EvolvR activity is mostly confined to the gene of interest (though not necessarily to the editing window), causing virtually no genomic mutations, only occasionally spilling over into other elements of the plasmid."*

Response to Referees

We thank the Editor and the Reviewers for their constructive feedback that enabled us (i) to enhance the clarity of our manuscript by refining its framing, and (ii) to improve its rigor via updated statistical analysis.

Reviewer #2

I appreciate the authors thoughtful responses, their additional experiments, and their dutiful follow-up on the questions I raised regarding intracellular plasmid variability. The authors have addressed the majority of my concerns. My primary concern is still regarding the framing of this paper. Figure 1 introduces the theme of this paper as intracellular plasmid variability and presents beautiful simulations to quantify the statistics of accumulation of mutation. As I see it (and I may have missed something), no other figure in the paper continues on this vein, but rather speaks towards general phenotypic outcomes of mutation using the EVOLVR system cleverly developed by the authors.

The supplementary figure 5 shows the outcomes of whole plasmid sequencing experiments on individual colonies, which are presumed to be isogenic variants in the traditional sense. However, the issue of intracellular plasmid variability as introduced in Figure 1 is still unaddressed, in this regard. A single cell can have multiple mutants of a plasmid and whole plasmid sequencing, NGS methods do not quantify this intrinsic genetic variation but rather present a measurement of a bulk sample of hundreds of thousands of cells.

I agree with the authors that development of methodologies to measure such intrinsic variability would be outside the scope of this work, but I think Figure 1 is quite confusing, in that it sets the expectation that the focus of this work will be on that theme. The subsequent figures are fine and tell a strong story, but the framing of the paper just needs some adjustment. No additional experiments are required, as I can tell.

To clarify the framing of the paper, we included the following right after presenting the simulation data in Fig. 1: *“In what follows, we consider the population-level effects of plasmid mutations, emphasizing how circuit-level properties (e.g., PCN, type and location of the mutation, regulatory architecture) modulate the phenotypic mutation rate within a population. While not the focus of our study, it is important to note that the co-occurrence of multiple plasmid variants within a single cell can result in sophisticated*

dynamics due to the interplay of evolutionary forces acting at two scales: in addition to competition between cells, host fitness is also impacted by replication interference between plasmids harbored within the same host. Studying this latter, intracellular competition, is possible via novel experimental tools such as engineered dimer plasmids (51) or via Cas9-based lineage recording (52) to quantify the fundamental trade-off between the two levels of selection, but falls outside the scope of our work.” This way we (i) emphasize that the present work focuses on population-level phenotypic effects of plasmid mutations, which are first revealed computationally and then subsequently confirmed experimentally; and (ii) acknowledge that high PCN can introduce competition both within and across cells, but addressing this lies beyond the scope of our work. We believe that this revision eliminates the source of confusion identified by the Reviewer and provides a clear articulation of the manuscript’s central focus.

One minor comment: In Supplementary Figure 5 (and elsewhere throughout the paper), standard error is used to plot the error bar. This is misleading, as the standard error represents the standard deviation normalized by an additional sampling factor n — (\sqrt{n} to be precise). I recommend the authors use 95% confidence intervals instead, as standard deviation is often too generous of a measure while standard error is too conservative. Using a confidence interval is generally statistically more rigorous, as well.

Following this helpful recommendation, we updated the statistical analysis so that error bars correspond to the 95% confidence intervals throughout both the manuscript and the Supplementary Information, except for the boxplot figures where the lower and upper bounds of the box represent the first quartile (Q1) and third quartile (Q3), respectively. With $IQR = Q3 - Q1$ denoting the interquartile range, the whiskers extend from the box to the minimum and maximum non-outlier values such that the lower and upper thresholds are defined as $Q1 - 1.5IQR$ and $Q3 + 1.5IQR$, respectively. This is now explicitly mentioned in the Methods as well as in the figure captions.

Reviewer #3

The authors have thoroughly and convincingly addressed my comments and concerns. I recommend this study for publication in Nature Communications.

We thank the Reviewer for their constructive feedback and suggestions that helped us not only improve the results presented in the original submission, but also sparked novel lines of research inquiry significantly expanding the scope of our work throughout the review process.